# Estimates of the prevalence of male circumcision in sub-Saharan Africa from 2010–2023 —A systematic review and meta-analysis

Basant Elsayed[1], Mohamed Elmarasi[1], Ruvarashe J. Madzime[2], Lovemore Mapahla[2], Menatalla M. S. ElBadway[1], Tawanda Chivese[1]*

1 Department of Population Medicine, College of Medicine, QU Health, Qatar University, Qatar,
2 Department of Global Health, Division of Epidemiology and Biostatistics, Faculty of Medicine and Health Sciences, Stellenbosch University, Cape Town, South Africa

☯ These authors contributed equally to this work.
* tchivese@gmail.com

**Data Availability Statement:** All relevant data are within the manuscript and its Supporting Information files.

## Abstract

### Background

Male circumcision (MC) is a key part of the package of interventions to prevent HIV, the biggest health challenge in sub-Saharan Africa.

### Objective

To estimate the male circumcision prevalence and to evaluate the progress towards meeting WHO targets in sub-Saharan Africa during the period 2010–2023.

### Methods

We carried out a systematic review and meta-analysis of studies published during the period 2010–2023. We searched PubMed, Scopus, Cochrane CENTRAL, Google Scholar, WHO and the Demographic and Health Survey for reports on MC prevalence in sub-Saharan Africa. MC prevalence was synthesized using inverse-variance heterogeneity models, heterogeneity using $I^2$ statistics and publication bias using funnel plots.

### Results

A total of 53 studies were included. The overall prevalence during the study period was 45.9% (95% CI 32.3–59.8), with a higher MC prevalence in Eastern (69.9%, 95%CI 49.9–86.8) compared to Southern African (33.3%, 95%CI 21.7–46.2). The overall prevalence was higher in urban (45.3%, 95%CI 27.7–63.4) compared to rural settings (42.6%, 95% 26.5–59.5). Male circumcision prevalence increased from 40.2% (95% CI 25.0–56.3) during 2010–2015 to 56.2% (95% CI 31.5–79.5) during 2016–2023. Three countries exceeded 80% MC coverage, namely, Ethiopia, Kenya and Tanzania.

**Funding:** The author(s) received no specific funding for this work.

**Competing interests:** The authors have declared that no competing interests exist.

## Conclusion

Overall, the current MC prevalence is below 50%, with higher prevalence in Eastern African countries and substantially lower prevalence in Southern Africa. Most of the priority countries need to do more to scale up medical male circumcision programs.

## Introduction

The HIV epidemic remains the biggest health challenge facing sub-Saharan Africa, with an estimated 38.4 million people living with HIV, 1.5 million new infections in 2021 and 650 000 deaths from HIV-related illnesses [1, 2]. Male circumcision, a low cost, minimally invasive and, in many settings, culturally acceptable procedure, has been shown to be effective in reducing the sexual transmission of the human immunodeficiency virus (HIV) and other sexually transmitted infections (STIs) [3]. Systematic reviews and meta-analyses of both randomized controlled trials and observational studies have consistently shown that male circumcision reduces the risk of HIV acquisition by between 40% and 60% [4–6]. Male circumcision has also been associated with a reduced risk of HIV and STIs in partners of circumcised men [5]. Consequently, since 2007, voluntary medical male circumcision (VMMC) has been recommended by the World Health Organization (WHO) and the Joint United Nations Programme on HIV/AIDS (UNAIDS) as part of a package of HIV prevention strategies, alongside other behavioural, structural and pharmacological HIV prevention strategies [6]. In sub-Saharan Africa, this has resulted in a gradual increase in the prevalence of VMMC, paralleled with a gradual decrease is HIV prevalence due to multifactorial interventions [7]. However, the progress towards reaching enough VMMC coverage in men aged 15–49 years has been uneven and data are not readily available about the current prevalence of male circumcision in these countries [7].

Although highly efficacious antiretroviral drugs are now widely accessible to people living with HIV, 470 000 people died from HIV-related illnesses in 2018 [8]. Being cognisant of the disproportionate burden of HIV in sub-Saharan Africa, the WHO prioritized 15 countries in Eastern and Southern Africa for support in setting up and scaling up VMMC, in order to increase male circumcision coverage in their countries [9]. These countries, have a high prevalence of HIV and traditionally low male circumcision prevalence, and include ten countries in Southern Africa, which are South Africa, Zambia and Botswana and four in Eastern Africa, namely, Kenya, Tanzania and Uganda [10]. Notably, modelling data had suggested that increasing VMMC coverage to 80% would result in a reduction of 3.4 million new cases of HIV [11]. Coupled with other interventions, VMMC could potentially help in ending the HIV epidemic in the sub-Saharan African region.

Globally the prevalence of male circumcision is estimated to be between 36.7%– 38.7%, with most of male circumcision being done for either religious or cultural reasons, for example Jewish and Muslim populations are likely to have a circumcision prevalence near 100% [12]. In sub-Saharan Africa, the countries with the highest male circumcision proportions are mainly Muslim majority countries [13]. However, around the year 2010, other countries began to offer VMMC as part of a suite of HIV prevention strategies, driven by multilateral support from organisations such as the WHO, the UNAIDS and the US President's Emergency Plan for AIDS Relief (PEPFAR) [14]. However, the UNAIDS Fast-Track target for VMMC that 90% of males aged 10–29 years will have received VMMC services by 2021 in priority settings in sub-Saharan Africa, is yet to be achieved in many of the priority countries [15]. Although male

circumcision prevalence and uptake statistics are sometimes reported in some countries, these data are not reported in a uniform pattern and data on the current progress in male circumcision coverage in the 15 priority countries are not readily available. The aim of this systematic review was to assess the prevalence of male circumcision in the 15 priority countries in sub-Saharan Africa during the period from 2010 to 2023. Further, we investigated the change in male circumcision prevalence during the same period and assessed the progress towards reaching of 80–90% male circumcision coverage in each country.

## Methods

### Study design and protocol registration

This study design was a systematic review and meta-analysis design and followed the Preferred Reporting Items for Systematic Reviews and Meta-Analysis (PRISMA) guidelines [16]. The study protocol was registered in International prospective register of systematic reviews (PROSPERO, ref—CRD42021271042).

### Data sources

A comprehensive search for data sources was carried out in PubMed, Scopus and Google Scholar without language filters. The initial search was carried out on 25th October 2021 and an updated search in April 2023 using each country separately. We also searched for data sources for each country on the Google Search Engine, and websites of the Demographic and Health Survey (DHS), and the WHO.

### Search strategy

A comprehensive search strategy using a filter for sub-Saharan Africa was used to identify eligible data sources. This search strategy was based on the following key concepts: male circumcision using MeSH terms and keywords (medical circumcision, traditional circumcision, circumcision among adolescent boys, rite of passage, infant male circumcision, thematic synthesis male circumcision, traditional vs medical circumcision) and African countries (Botswana, Eswatini, Ethiopia, Kenya, Lesotho, Malawi, Mozambique, Namibia, Rwanda, South Africa, South Sudan, Tanzania, Uganda, Zambia, Zimbabwe). We also searched for countries with more than one name or that changed their official name in the last ten years. For example, Swaziland changed to Eswatini, and therefore both names were used in searches.

### Screening of studies for inclusion

The records of the identified data sources across the search databases were exported into Endnote for deduplication and then to Rayyan [https://rayyan.qcri.org/], a systematic review management software, for screening. Initial screening of the studies using the titles and the abstracts was done for preliminary inclusion by two authors (BE and ME) per data source. Where conflicts in the selection of studies appeared, a third author (TC) was consulted. Subsequently, two reviewers independently assessed each of the full text of studies eligible for full-text screening. A predefined eligibility criteria checklist was used for the full-text assessment.

### Eligibility

**Inclusion criteria.** All cross-sectional, population-based and cohort studies that investigated the prevalence of male circumcision in sub-Saharan Africa were eligible for inclusion in this review. The eligible studies should have been published during the period 1 January 2010 and 7 April 2023. This period 2010–2023 was selected as most VMMC programs started either

just before or during the year 2010 (in the case of South Africa) after the WHO recommendation in 2007 [14]. No language restriction was applied, and we did not attempt to distinguish between VMMC and traditional and cultural male circumcision as many data sources did not do separate analyses by type of male circumcision and others did not specify the type of circumcision.

**Exclusion criteria.** Duplicate data sources, studies that did not report male circumcision prevalence by country in the analysis, studies that were published before or after the period of interest and studies that included data for countries outside the 15 priority countries were excluded.

## Key definitions

Male circumcision was defined as the complete removal of the foreskin from a penis. However, this was self-reported in many studies, so we accepted the data sources' definitions of male circumcision. To estimate the prevalence of male circumcision at a country level, we required nationally representative data and therefore attempted to distinguish between nationally representative data and local data. For this purpose, nationally representative studies were defined as studies that sampled from the whole country using a probability sampling method.

## Data extraction

From each study meeting the eligibility criteria, two independent reviewers (BE and ME) extracted data into a predefined extraction form in Microsoft Office Excel. The two investigators compared their findings and discussed to resolve any differences and consulted the third reviewer (TC) when they failed to reach consensus. Data extracted included authors, dates of data collection, country, study design, setting, age groups included, total sample size, number of participants with male circumcision, sampling method, and response rate. We extracted the male circumcision data within age-groups if the authors reported the age-specific data. One study, Tram, et al. 2014, reported findings of DHS surveys carried out in 12 countries during the period 2006–2011 (24). However, if the DHS survey was published after 2010, data from Tram 2014 were not used, rather the original DHS report was then used. This was done for the Uganda DHS 2011 survey, Ethiopia DHS 2011, Lesotho DHS 2009, Malawi DHS 2010, Kenya DHS 2009, Rwanda DHS 2010, Mozambique DHS 2011, Tanzania DHS 2011, and the Zimbabwe DHS 2007.

## Quality appraisal

Two reviewers independently assessed the quality of the studies using the risk of bias checklist for prevalence studies described by Hoy et al. [17]. The tool uses a ten-item rating scale to assess the internal (items 1–4), and external validity (items 5–11) and the tenth item is a summary of the overall risk of study bias. Each item was assigned a score of 0 [yes] or 1 [no], and scores were summed across items to generate an overall quality score that ranged from 0 to 9. The risk of bias in each study was rated as low (0–3), moderate (4–6), or high (7–9) risk dependent on the number of questions answered as "yes [low risk]".

## Data synthesis

We described the characteristics of the included studies using a table and narratively in the text. We recalculated estimates of the prevalence of circumcision among men (number of cases/ sample size) based on the information on crude numerators and denominators provided in the individual studies. For each country, the prevalence of male circumcision was estimated

using the highest ranked representative study that was available, and in this case, most of these were DHS surveys. If more than one country-level representative study was available for a specific time period, a meta-analysis was carried out.

Where meta-analysis was possible, we pooled prevalence from individual studies using a bias adjusted inverse variance heterogeneity model [18]. We used the Freeman-Turkey double arcsine transformation to stabilize the variance before pooling the studies. We assessed heterogeneity of studies using Cochran's Q test, with a cut-off of 0.1, and the $I^2$ statistic. Cochran's Q test a generates a probability that indicates the consistent variation across studies rather than within subjects within a study with the null hypothesis assuming that male circumcision is the same across studies and variations are simply caused by chance. We interpreted $I^2$ values over 50% to indicate substantial heterogeneity and above 75% high heterogeneity. If there was substantial heterogeneity, subgroup analysis was carried out. Preplanned subgroup analyses were done by age-group, by rural/urban settings, time period (before and after 2015) and by subregion (Eastern and Southern regions). Maps of the prevalence of male circumcision in each country before and after the year 2015 were generated using Tableau^TM software (https://www.tableau.com/en-gb/academic/students). Publication bias was assessed using funnel plots, Doi plots [19] and Eiger's p-value. We reported prevalence and their 95% confidence intervals [95%CI] and used forest plots to display the findings. We used the metan package in Stata version 17 [26] for all meta-analyses.

## Ethics

Given that this was a study for a systematic review which utilised published data, ethical approval was not required.

## Results

### Search results

We identified 395 records from all searches and 285 records were assessed for inclusion using the title and abstract only, after the removal of duplicates (Fig 1). After exclusion of 210 irrelevant records, the full texts of 75 studies were screened, and 22 data sources were excluded. A total of 53 data sources were finally included.

### Characteristics of included studies

The 53 included studies had a combined total of 264, 110 men and were from 14 countries (S1 Table). The sample size ranged from 170 in a study from Tanzania to 36,628 in a study done in South Africa [20, 21]. Most (n = 44) of the studies were cross-sectional studies, three studies were cohort studies, one prospective observational study, and one pair-matched community-randomized trial. All the countries had data from at least one of the DHS. There were no data sources from South Sudan. We identified 36 studies which had surveys done before 2015 while 17 were done after 2015. One study reported the male circumcision prevalence among 12 countries in the sub-Saharan Africa during different surveys in men aged above 15 years [22].

### Risk of bias assessment of included studies

The majority (n = 42) of the studies had a total score which reflected a low risk of bias, and the remaining (n = 11) studies had a total score which reflected a moderate risk of bias. However, there were deficiencies in the definition of male circumcision with 40 studies not giving an

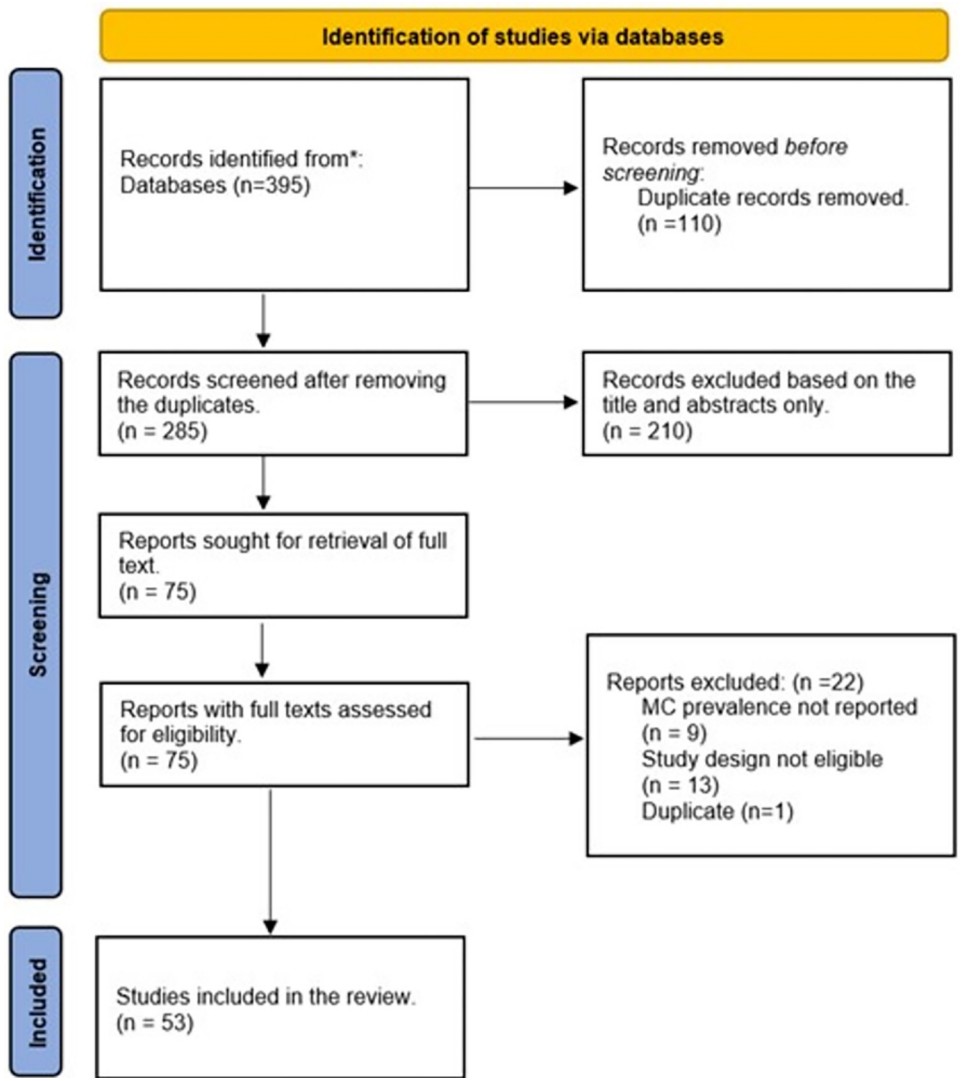

**Fig 1. PRISMA flow chart.** The figure shows the search process. Abbreviations: MC–male circumcision.

acceptable case definition of male circumcision. Similarly, item 7 was deficient in 41 studies where circumcision was measured by self-report (S2 Table).

## Prevalence of male circumcision during 2010–2023

The raw prevalence of male circumcision ranged from 4.6% in a study from South Africa in 2011 to 98.8% in a study from Tanzania in 2011 [20, 23]. The overall prevalence during the study period was 45.9% (95% CI 32.2–59.8) with high heterogeneity ($I^2$ = 100%) (Fig 2). The Doi plot showed no asymmetry suggesting no evidence of publication bias (S1 Fig). There was significant jump in the overall male circumcision prevalence from 40.2% (95% CI 25.0–56.3) during 2010–2015 to 56.2% (95% CI 31.5–79.5) during 2016–2023 (Fig 2).The prevalence of male circumcision was higher in urban settings (50.9%, 95% CI 35.6–66.2, $I^2$ = 99.8%) compared to studies in the mixed urban/rural (43.8%, 95% CI 34.4–53.3, $I^2$ = 100%) and purely rural settings (26.2%, 95% CI 16.0–37.8, $I^2$ = 99.8%), with significant evidence of

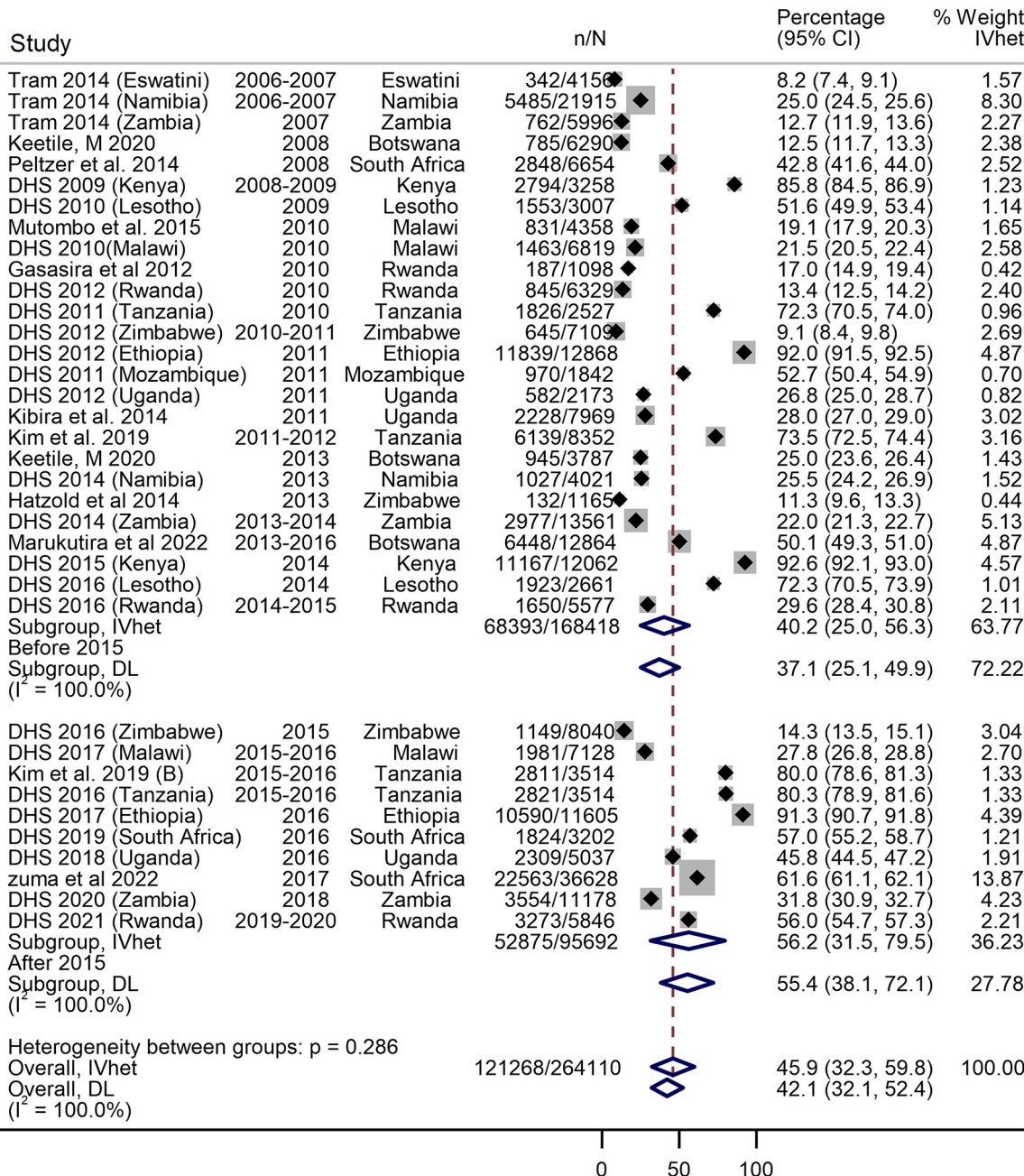

**Fig 2. Forest plot showing the overall prevalence of male circumcision during 2010–2023.** The forest plot includes data from nationally representative studies only.

subgroup interaction (p = 0.031) (S3 Table). Male circumcision prevalence was higher in the Eastern African countries (69.8%, 95% CI 49.8–86.8, $I^2$ = 100%) than the prevalence in the Southern African sub-region (33.4%, 95% CI 21.7–46.2, $I^2$ = 99.9%) (S4 Table). The prevalence of male circumcision was highest in the 30–39 years age group (50.0%, 95% CI 30.9–69.0, $I^2$ = 99.9%) and the 40–49 years age group (49.0%, 95% CI 29.2–69.0, $I^2$ = 99.8%) (S5 Table).

## Prevalence of male circumcision in the 14 countries

**Eastern Africa.** Figs 3 and 5 show changes in the prevalence of male circumcision in Eastern African countries during the study period.

In Kenya, nationally representative studies showed a consistently high prevalence of male circumcision during the study period. Results of a DHS survey showed a male circumcision prevalence of 85.9% (95% CI 84.7–87.0) among men aged 15 years and above before 2010 [22]. Another study carried out in nine regions showed a high prevalence of male circumcision prevalence among Kenyan HIV-uninfected men aged 15–64, of 91.1% (95% CI 90.4–91.8) pre-2010 and 85.8% (95% CI 85.0–86.5) in 2012 [24]. The Kenya DHS in 2014 reported an male circumcision prevalence of 93% (95% CI 92.5–93.4) [25]. However, some studies reported a lower prevalence of male circumcision from certain regions. In Kisumu, among men aged 15–49, three consecutive surveys [26] showed a male circumcision prevalence of 31.9 (95% CI 28.4–35.5) in 2009, 48.8% (95% CI 46.2–51.4) in 2011 and 59.7% (95% CI 57.0–62.3) in 2013. The remaining two studies reported a male circumcision prevalence of 50.6% (95% CI 49.3–51.9) in 2014 to 71.2% (95% CI 69.7–72.7) in 2019 in Siaya, Kisumu, Homa Bay and Migori Counties among men younger than 35 years [27, 28].

In Tanzania, nationally representative studies showed an increase in the male circumcision prevalence from 73.5% (95% CI 72.5–74.4) during 2011–2012 to 80% (95% CI 78.6–81.3) during 2015–2016 [29–32]. The highest male circumcision prevalence reported in Tanzania was 98.8% (95% CI 95.8–99.7), although this was in Northen Tanzanian communities which are traditionally circumcising [20].

In Uganda, a DHS survey in 2010 showed a male circumcision prevalence of 27% (95% CI 25.2–28.9) [22], which was similar to estimates from the Uganda AIDS Indicator Survey of 26.7% (95% CI 25.8–27.6) in 2011 [33–35]. Surveys done in 2015 and 2016 showed a slight increase in the male circumcision prevalence to 31.3 (95% CI 26.0–36.0) and 45.2 (95% CI 43.9–46.5) respectively [36, 37]. Circumcision prevalence was low among non-Muslim men, for example, in Rakai, the prevalence of male circumcision was 28.8% (95% CI 27.7–29.0) during 2010–2011 [38].

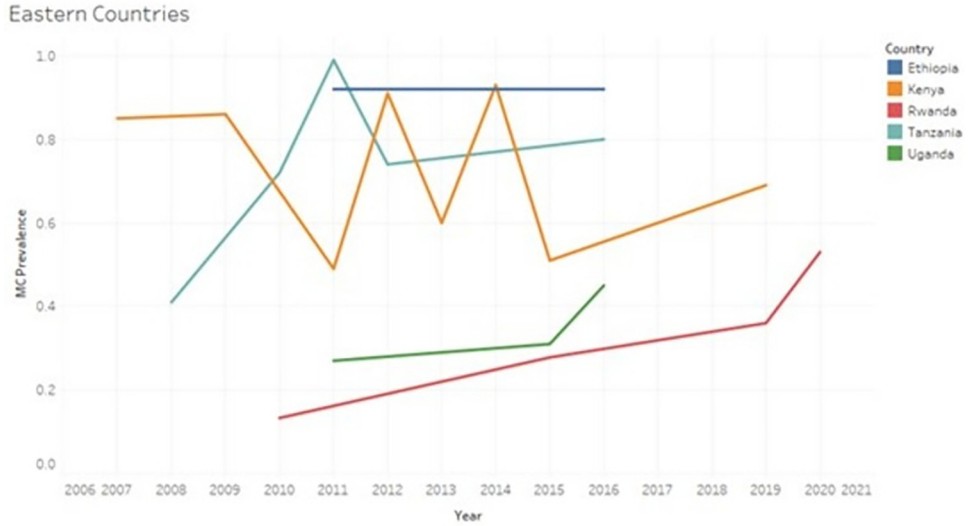

**Fig 3. Changes in the prevalence of male circumcision in Eastern African countries during the period 2010–2023.** The graph shows the changes in the prevalence of male circumcision in Eastern African countries during the period 2010–2023.

In Ethiopia, three nationally representative studies were carried out, and they showed a male circumcision prevalence of 92% (95% CI 91.5–92.4) in 2011 [39, 40], 92% (95% CI 91.7–92.6) in 2011 and 91.2% (95% CI 90.7–91.7) in 2016 in men aged 15 years and above [22, 39, 40]. The lowest reported male circumcision prevalence was 61.2% (95% CI 57.9–64.4) from a local study in the Gambella region in 2016 [41].

In Rwanda, three consecutive demographic and health surveys showed an increase in the male circumcision coverage among males aged 15–59 years. The male circumcision prevalence in the DHS in 2010, 2014–2015 and 2019–2020 was 13.3% (95% CI 12.5–14.2), 27.8% (95% CI 26.7–28.9) and 52.5% (95% CI 51.3–53.7) respectively [42–44]. Local studies also showed gradual increase in male circumcision prevalence, for example, in Nyanza district, male circumcision prevalence increased from 17% (95% CI 14.9–19.4) in 2010 to 35.8% (95% CI 31.5–40.4) in 2019 [45, 46].

**Southern Africa.** Figs 4 and 5 show the changes in the prevalence of male circumcision in Southern African countries during the study period.

In South Africa, three nationally representative studies were carried out, and they showed a gradual rise in male circumcision prevalence from 42.8% (95% CI 41.6–44.0) in 2008 [47], 55.6% (95% CI 54.0–57.2) in 2016 [48] and 61.6% (95% CI 61.1–62.1) in 2017 [21]. The remaining 4 regional and local studies showed a male circumcision prevalence which ranged from 4.6% in rural KwaZulu-Natal [49] during 2011 to a high of 56.7% in Orange Farm in 2017 [50] (S4 Table).

In Zimbabwe, four nationally representative studies were carried out, and they showed a male circumcision prevalence of 9% (95% CI 8.4–9.7) and 9.2% (95% CI 8.6–9.9) in 2010–2011 [22, 51], 11.3 (95% CI 9.6–13.3) [52] in 2013 and 14.3% (95% CI 13.6–15.1) [53] in 2015. In two local Zimbabwean provinces, the male circumcision prevalence was 20.3% (95% CI 17.9–22.9) in 2009 [54]. In the Mazowe District, the male circumcision prevalence was 15.3% (95% CI 11.7–19.8) among men aged 18–49 in 2014 [55]. Nationally representative data were not available from the period 2015–2023 (Fig 3).

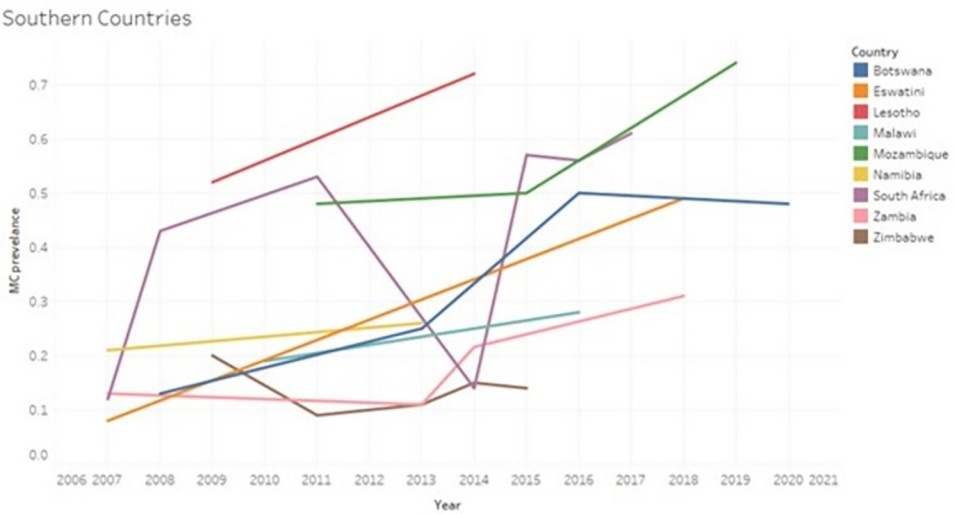

**Fig 4. Changes in the prevalence of male circumcision in Southern African countries during the period 2010–2023.** The graph shows the changes in the prevalence of male circumcision in Southern African countries during the period 2010–2023.

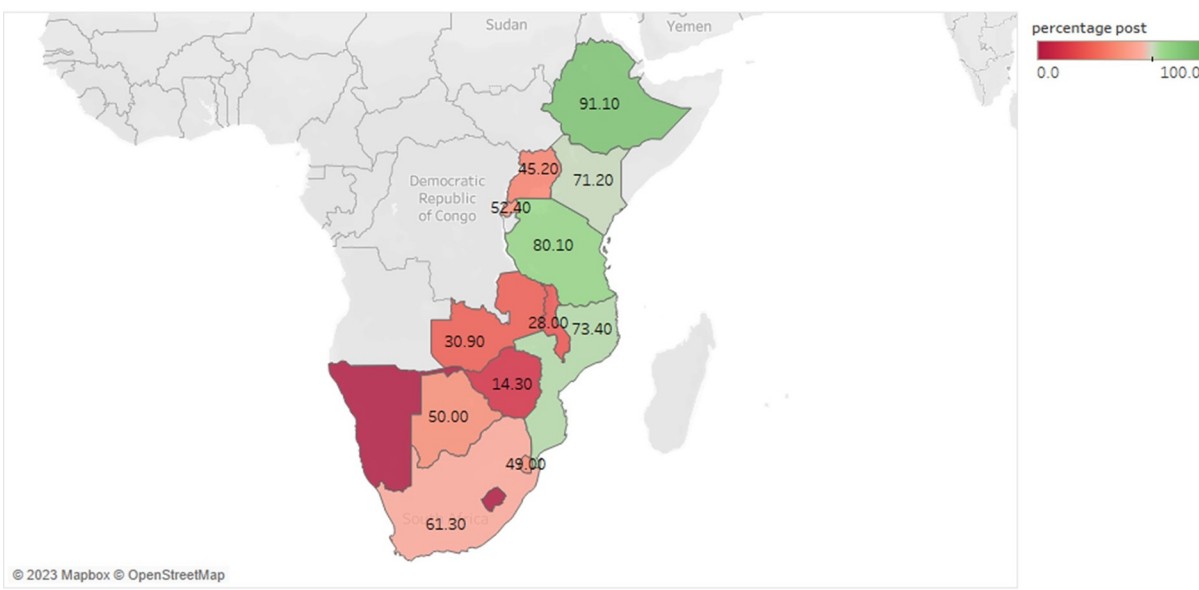

**Fig 5. Maps showing changes in the prevalence of male circumcision in Eastern African countries during 2010–2015 and 2016–2023.**
Maps showing the prevalence of male circumcision during 2010–2015 and 2016–2023 among sub-Saharan Africa countries.

In Malawi, three nationally representative studies were carried out, and they showed a gradual increase in the male circumcision prevalence of 19.1% (95% CI 17.9–20.3) and 22% (95% CI 21.1–23.0) [22, 56] in 2010 and 28% (95% CI 27.1–29.0) [57] during 2015–2016.

In Zambia, nationally representative studies suggested a gradual increase in the male circumcision prevalence from before 2010 (2007: 12.8%, 95% CI 12.0–13.6), to 11.4% (95% CI 10.2–12.8) during 2023 [22, 58], 21.6 (95% CI 20.9–22.3) during 2013–2014 to 30.9% (95% CI 30.1–31.7) during 2018 [59, 60].

In Botswana, one study reported two national-wide consecutive surveys done in 2008 and 2013 that showed an increase in the male circumcision prevalence from 12.5% (95% CI 11.7–13.3) to 25.2% (95% CI 23.9–26.7) respectively [61]. Another nationally representative study reported male circumcision prevalence of 50.1% (95% CI 49.3–51.0) during the period 2013–2016 [62]. The latest survey done in 2020 among undergraduate male students aged 17–25 showed similar findings (47.9%, 95% CI 42.3–53.5) [63].

In Eswatini, one nationally representative study reported a male circumcision coverage of 8.2% (95% CI 7.4–9.1) in 2006–2007 (24), while a survey done in Shiselweni region reported a male circumcision prevalence of 49.4% (95% CI 44.6–54.2) in 2018 (Shezi et al 2023) [22, 64].

In Mozambique, one nationally representative study reported a male circumcision prevalence of 48% (95% CI 46.5–49.5) in men above 15 years in 2011 [22]. In Chókwè District, the male circumcision prevalence increased from 50% (95% CI 47.1–53) in 2014–2015 to 73.5 (95% CI 70.9–75.9) [65].

In Lesotho, the male circumcision prevalence increased from 5.3 (95% CI 4.6–6.1) in 2009 to 72% (95% CI 70.3–73.6) in 2014 according to data from the Lesotho DHS [22, 66].

In Namibia, the male circumcision prevalence increased slightly from 21% (95% CI 19.7–22.3) in 2006–2007 to 25.5% (95% CI 24.2–26.9) in 2013 according to data from two DHS surveys [22, 67].

The pooled prevalence of male circumcision in each country is shown in S6 Table. Fig 5 shows the overall prevalence of male circumcision in each country during 2010–2015 and 2016–2023.

## Discussion

In this systemic review and meta-analysis of 53 studies from 14 of the 15 HIV priority countries, we found that overall prevalence of male circumcision during 2010–23 was 46%. The prevalence of male circumcision increased from 40% during 2010–2015 to 56% during 2016–2023 in the priority countries. Further, the prevalence of male circumcision was higher in Eastern African countries compared to Southern African countries.

We found that during the study period, overall, just under half, 46%, of men of reproductive age in the priority African countries were circumcised, with variation between countries and between regions. Previous estimates from a published study have suggested a global male circumcision prevalence of 36.7% in 2015 [12]. Notably, current estimates of the male circumcision prevalence in the region have not been reported in reviews. In addition, our findings confirm the disparity in male circumcision prevalence in rural and urban regions, findings which are in concurrence with the primary studies that we included [25, 31, 40, 66]. Our findings suggest a need to up-scale the male circumcision uptake in the priority countries.

Our analyses showed that the prevalence of male circumcision increased by 16 percentage points from 40% during 2010–2015 to 56% during 2016–2023. This suggests that the promotion of VMMC has an impact, and that this impact could be better. However, there are also some notable exceptions to this trend. For example, some countries with high HIV prevalence, such as Lesotho and Eswatini, have reported very low rates of circumcision, 5.3% (95% CI 4.6–6.1), and 8.2% (95% CI 7.4–9.1) respectively [22]. This highlights the importance of understanding the cultural and social factors that influence circumcision practices in different populations.

Our findings suggest that only three countries, Ethiopia, Kenya, and Tanzania, have reached the WHO target of 80% in at least one of the surveys done in these countries. The current study also showed that the prevalence of male circumcision was still low in Southern Africa countries which have the highest burden of HIV. The highest reported prevalence in South Africa was 56.7%, in Botswana, 47.9%, in Eswatini, 26% [68], in Namibia, 21%, and 5.3% in Lesotho. A potential explanation for the differences in VMMC prevalence between Eastern and Southern African countries is related to cultural and religious practices towards circumcision. In Kenya, Rwanda, Tanzania and Uganda, significant proportion of the population are of Islamic religion where circumcision is commonly performed at birth [69]. On the other hand, circumcision practices in Southern Africa vary according to religion and local cultures the proportion of Muslim populations is relatively small [5]. Some local cultures practice circumcision as a rite of passage [70]. Although circumcision has a long history in many African cultures and is often seen as a rite of passage or a symbol of masculinity, attitudes towards circumcision vary widely within and between countries, and may be influenced by factors such as religion, ethnicity, and education [70]. Understanding these cultural and social factors is important in designing effective VMMC programs that are culturally sensitive and acceptable to target populations.

Out of the 14 countries, Ethiopia, Kenya and Tanzania had surpassed the WHO target of 80% during at least one survey. While the male circumcision target has not been met in the remaining countries, considerable progress was made with most of the priority countries more than doubling their male circumcision coverage. The reasons for the high male circumcision prevalence in the Eastern African countries have been discussed before. However, our findings suggest that interventions to increase male circumcision acceptability and coverage may have contributed to the observed increase in the male circumcision prevalence in all these countries. These interventions included awareness campaigns, and mobile circumcision clinics [12, 71, 72]. However, our results also suggest a need for continued investment to promote male circumcision uptake to reach, and possibly surpass, the WHO target. The benefits of this intervention cannot be understated, modelling data suggested that a quarter of a million new infections were prevented through VMMC in sub-Saharan Africa during the decade 2008–2018 [73]. Notably, male circumcision offers lifelong protection, and therefore has larger future benefits [73]. For example, if other behavioural and structural interventions remain constant, the mathematical models forecast that the impact of the VMMC up to 2018 alone would result in the prevention of 1.5 million cases of HIV by 2030 and that this number could triple to 4.5 million averted HIV cases by 2050 [73].

A strength of this study was that exhaustive search for data sources was carried out in both scientific databases and databases of government and inter-governmental organisations. The analysis used high quality data that were nationally representative, and therefore reflects the prevalence of male circumcision in these countries. Limitations of the current study include the lack of enough studies per country, the lack of data from South Sudan and the lack of recent data in most countries. Lastly, male circumcision data were not reported according to the type of circumcision, i.e. medical or customary circumcision.

## Conclusion

Overall, the current prevalence of male circumcision is just below 50%, with higher prevalence observed in Eastern African countries and substantially lower prevalence in Southern Africa. While some progress in the male circumcision coverage was observed during the study period, most of the priority countries still need to scale up male circumcision programs, especially in rural regions. Further, countries should consider prioritizing scaling up of male circumcision

in older males (>50 years) as the prevalence of male circumcision in this demographic lags behind that in other age groups.

## Supporting information

**S1 Checklist. PRISMA 2020 checklist: The PRISMA checklist for systematic reviews and meta-analyses.**
(DOCX)

**S1 Table. Characteristics of included studies.** This table shows the characteristics of included studies.
(DOCX)

**S2 Table. Assessment of risk of bias of included studies using Hoy et al 2012.** This table shows the assessment of risk of bias of the included studies using 11 item Hoy et al tool.
(DOCX)

**S3 Table. Male circumcision prevalence in urban vs rural settings.** This table shows the prevalence of male circumcision in sub-Saharan African countries based on the rural and urban settings.
(DOCX)

**S4 Table. Male circumcision prevalence in eastern and southern regions.** This table shows the prevalence of male circumcision in Sub-Saharan African countries based on the region.
(DOCX)

**S5 Table. Male circumcision prevalence in different age groups.** This table shows the prevalence of male circumcision in Sub-Saharan Africa among different age groups.
(DOCX)

**S6 Table. Male circumcision prevalence by country.** This table shows the prevalence of male circumcision in Sub-Saharan African countries for each country.
(DOCX)

**S7 Table. Male circumcision prevalence by time period.** This table shows the prevalence of male circumcision in Sub-Saharan African countries by the time period: before 2015 and after 2015.
(DOCX)

**S1 Fig. Doi plot to assess publication bias.** This figure shows the Doi plot used to assess the publication bias. The Doi plot showed no asymmetry suggesting no evidence of publication bias.
(PNG)

## Author Contributions

**Conceptualization:** Basant Elsayed, Mohamed Elmarasi, Ruvarashe J. Madzime, Lovemore Mapahla, Tawanda Chivese.

**Data curation:** Basant Elsayed, Mohamed Elmarasi, Ruvarashe J. Madzime, Lovemore Mapahla, Menatalla M. S. ElBadway, Tawanda Chivese.

**Formal analysis:** Basant Elsayed, Mohamed Elmarasi, Ruvarashe J. Madzime, Lovemore Mapahla, Menatalla M. S. ElBadway, Tawanda Chivese.

**Investigation:** Basant Elsayed, Mohamed Elmarasi, Ruvarashe J. Madzime, Lovemore Mapahla, Menatalla M. S. ElBadway, Tawanda Chivese.

**Methodology:** Basant Elsayed, Mohamed Elmarasi, Ruvarashe J. Madzime, Lovemore Mapahla, Tawanda Chivese.

**Project administration:** Basant Elsayed, Mohamed Elmarasi, Tawanda Chivese.

**Resources:** Basant Elsayed, Mohamed Elmarasi, Tawanda Chivese.

**Software:** Basant Elsayed, Mohamed Elmarasi, Tawanda Chivese.

**Supervision:** Tawanda Chivese.

**Validation:** Basant Elsayed, Mohamed Elmarasi, Tawanda Chivese.

**Visualization:** Basant Elsayed, Mohamed Elmarasi, Tawanda Chivese.

**Writing – original draft:** Basant Elsayed, Mohamed Elmarasi, Tawanda Chivese.

**Writing – review & editing:** Basant Elsayed, Mohamed Elmarasi, Ruvarashe J. Madzime, Lovemore Mapahla, Menatalla M. S. ElBadway, Tawanda Chivese.

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
