## [Decision Letter · Decision Letter 0]

21 Dec 2023

PONE-D-23-37643

Estimates of the prevalence of male circumcision in Sub-Saharan Africa from 2010 – 2023 – a systematic review and meta-analysis

PLOS ONE

Dear Dr. Chivese,

Thank you for submitting your manuscript to PLOS ONE. After careful consideration, we feel that it has merit but does not fully meet PLOS ONE’s publication criteria as it currently stands. Therefore, we invite you to submit a revised version of the manuscript that addresses the points raised during the review process.

We look forward to receiving your revised manuscript.

Kind regards,

Taurayi A Tafuma, MPH, MBChB

Academic Editor

PLOS ONE

3. We note that  Figure 5 in your submission contain [map/satellite] images which may be copyrighted. All PLOS content is published under the Creative Commons Attribution License (CC BY 4.0), which means that the manuscript, images, and Supporting Information files will be freely available online, and any third party is permitted to access, download, copy, distribute, and use these materials in any way, even commercially, with proper attribution. For these reasons, we cannot publish previously copyrighted maps or satellite images created using proprietary data, such as Google software (Google Maps, Street View, and Earth). For more information, see our copyright guidelines: http://journals.plos.org/plosone/s/licenses-and-copyright.

a. You may seek permission from the original copyright holder of  Figure 5 to publish the content specifically under the CC BY 4.0 license.  

Reviewers' comments:

Reviewer's Responses to Questions

**Comments to the Author**

1. Is the manuscript technically sound, and do the data support the conclusions?

Reviewer #1: Yes

Reviewer #2: Yes

 **********

2. Has the statistical analysis been performed appropriately and rigorously?

Reviewer #1: Yes

Reviewer #2: Yes

 **********

3. Have the authors made all data underlying the findings in their manuscript fully available?

Reviewer #1: Yes

Reviewer #2: Yes

 **********

4. Is the manuscript presented in an intelligible fashion and written in standard English?

Reviewer #1: Yes

Reviewer #2: Yes

 **********

5. Review Comments to the Author

Reviewer #1: The title is well structured and aligned to the objectives, findings and conclusions

In most cases, the use of Sub-Saharan Africa in between a sentence need to be observed as it should be sub-Saharan Africa

On introduction, authors should give also HIV prevalence change in countries as they give the background of VMMC prevalence so that readers are directed on VMMC contribution to HIV prevalence. On this part on prevalence for 2021 was given yet the study is for 2010 to 3023.

On study design, first statement, write in past tense as it is in present tense

On page 6 and afterwards, seems font type and size changed, so check on this.

On analysis page 13, state the assumptions of Cochran's Q test so that its linked to the study an the test is usually used to test consistent of outcomes within different groups

On sample size, 53 records were finally used out of the initial 285 meaning about 3-4 records per country, are these suffice to represent the country and make conclusions about VMMC for a country.

In discussion, relate VMMC prevalence to HIV prevalence among men so that this will strengthen recommendations on scaling up VMMC which is below 50% in some countries.

In conclusion, relate recommendations to specific age groups as outlined in the results so that it will be more targeted on the age groups trailing behind.

Reviewer #2: The manuscript is technically sound, is presented in a clear and intelligible manner, and the data generally support the conclusions. However, I have queries about the following points/issues:

1. There is no clear rationale or justification provided for the time period selected (2010 to 2023) for inclusion of studies for review. How was the time period for studies to include determined? Was it an arbitrary decision that had no obvious justification or reasoning behind it? How does the time period selected for inclusion of studies assist or contribute towards achieving the objectives of this work? Could the rationale or justification for the time period selected (2010 to 2023) for inclusion of studies please be clearly stated in the manuscript/paper?

2. Search results initially yielded 395 records or studies, and then after assessment, some studies were excluded and a total of 53 data sources were finally included. Although there is a section of the paper titled "Characteristics of included studies"; this section only describes some features of studies that were included, it does not expressly and explicitly state why they were included; ie. inclusion criteria for inclusion of studies is not clearly stated. There is no section of the document that describes or categorically states the exclusion criteria that were used to make decisions to exclude those studies that were excluded; i.e. exclusion criteria for exclusion of studies is also not clearly stated. Would it not be better if there could be a section of the document that clearly and explicitly states and provides the inclusion and exclusion criteria for selecting studies/data sources for this work?

My recommendation is that the paper would be acceptable for publication after minor revisions to address and attend to the queries that I have raised above.

 **********

6. PLOS authors have the option to publish the peer review history of their article (what does this mean?). If published, this will include your full peer review and any attached files.

**Do you want your identity to be public for this peer review?** For information about this choice, including consent withdrawal, please see our Privacy Policy.

Reviewer #1: **Yes: **Fungai Hamilton Mudzengerere

Reviewer #2: **Yes: **Dr. Tonderai Kasu

 **********

---

## [Author Response · Author response to Decision Letter 0]

7 Jan 2024

Response: Thank you. The manuscript has been edited to meet the journal’s style requirements. 

Response: Thank you. The ethics statement only appears in the Methods section under “Ethics” and has been deleted from elsewhere. 

3. We note that Figure 5 in your submission contain [map/satellite] images which may be copyrighted. All PLOS content is published under the Creative Commons Attribution License (CC BY 4.0), which means that the manuscript, images, and Supporting Information files will be freely available online, and any third party is permitted to access, download, copy, distribute, and use these materials in any way, even commercially, with proper attribution. For these reasons, we cannot publish previously copyrighted maps or satellite images created using proprietary data, such as Google software (Google Maps, Street View, and Earth). For more information, see our copyright guidelines: http://journals.plos.org/plosone/s/licenses-and-copyright.

a. You may seek permission from the original copyright holder of Figure 5 to publish the content specifically under the CC BY 4.0 license. 

Response: Thank you – please note that this figure is not a copyrighted map but a map that we generated from the data synthesized in this meta-analysis. We have added the following statement in the methods, last paragraph on page 7

Maps of the prevalence of male circumcision in each country before and after the year 2015 were generated using TableauTM software (https://www.tableau.com/en-gb/academic/students).

Response: Thank you. Captions for the supporting information are added at the end of the manuscript. 

Response: Thank you. No retracted references are present in our reference list. 

5. Review Comments to the Author

Reviewer #1: 

The title is well structured and aligned to the objectives, findings and conclusions

In most cases, the use of Sub-Saharan Africa in between a sentence need to be observed as it should be sub-Saharan Africa

Response: Thank you for your comment. We have corrected this.

On introduction, authors should give also HIV prevalence change in countries as they give the background of VMMC prevalence so that readers are directed on VMMC contribution to HIV prevalence

Response: Thank you for your comment. We have added a sentence on the gradual decrease of HIV prevalence which is in parallel with a gradual increase in the coverage of VMMC in SSA, although it should be noted that the gradual decrease in HIV prevalence cannot be attributed to VMMC alone but to its combination with several multifactorial interventions.

“In sub-Saharan Africa, this has resulted in a gradual increase in the prevalence of VMMC, paralleled with a gradual decrease is HIV prevalence due to multifactorial interventions [8].”

On study design, first statement, write in past tense as it is in present tense

Response: Thank you for your comment. The sentence has been changed to the following: This study design was a systematic review and meta-analysis design that followed the Preferred Reporting Items for Systematic Reviews and Meta-Analysis (PRISMA) guidelines [16].

On page 6 and afterwards, seems font type and size changed, so check on this.

Response: Thank you for your comment. Consistent font type and size have now been used across the manuscript. 

On analysis page 13, state the assumptions of Cochran's Q test so that its linked to the study an the test is usually used to test consistent of outcomes within different groups

Response: Thank you for your comment. The following has been added to the statistical analysis paragraph: 

“Cochran’s Q test a generates a probability that indicates the consistent variation across studies rather than within subjects within a study with the null hypothesis assuming that male circumcision is the same across studies and variations are simply caused by chance.”

On sample size, 53 records were finally used out of the initial 285 meaning about 3-4 records per country, are these suffice to represent the country and make conclusions about VMMC for a country.

Response: Thank you for your comment. Some of these studies are nationally representative studies with rigorous design, and may suffice, with some limitations, in describing the prevalence of male circumcision in each country. We do agree that more studies would have provided better estimates and have also discussed this in the limitations. The local and regional studies would also not have been very useful in providing estimates. This is why we prioritized the nationally representative studies in the analyses. 

We have edited the discussion as follows:

Limitations of the current study include the lack of enough studies per country, the lack of data from South Sudan and the lack of recent data in most countries.

In discussion, relate VMMC prevalence to HIV prevalence among men so that this will strengthen recommendations on scaling up VMMC which is below 50% in some countries.

Response: Thank you for your comment. We have added the following: 

“The benefits of this intervention cannot be understated, modelling data suggested that a quarter of a million new infections were prevented through VMMC in sub-Saharan Africa during the decade 2008-2018 [74]. Notably, male circumcision offers lifelong protection, and therefore has larger future benefits [74]. For example, if other behavioral and structural interventions remain constant, the mathematical models forecast that the impact of the VMMC up to 2018 alone would result in the prevention of 1.5million cases of HIV by 2030 and that this number could triple to 4.5million averted HIV cases by 2050 [74].”

In conclusion, relate recommendations to specific age groups as outlined in the results so that it will be more targeted on the age groups trailing behind.

Response: Thank you for your comment. We have added the following:

“Further, countries should consider prioritizing scaling up of male circumcision in older males (>50 years) as the prevalence of male circumcision in this demographic lags behind that in other age groups.”

Reviewer #2: The manuscript is technically sound, is presented in a clear and intelligible manner, and the data generally support the conclusions. However, I have queries about the following points/issues:

1. There is no clear rationale or justification provided for the time period selected (2010 to 2023) for inclusion of studies for review. How was the time period for studies to include determined? Was it an arbitrary decision that had no obvious justification or reasoning behind it? How does the time period selected for inclusion of studies assist or contribute towards achieving the objectives of this work? Could the rationale or justification for the time period selected (2010 to 2023) for inclusion of studies please be clearly stated in the manuscript/paper?

Response: Thank you for your comment. Most VMMC programs started around the year 2010 after the WHO recommendation. We have added the following sentence:

“This period 2010-2023 was selected as most VMMC programs started either just before or during the year 2010 (in the case of South Africa) after the WHO recommendation in 2007 [14].”

2. Search results initially yielded 395 records or studies, and then after assessment, some studies were excluded and a total of 53 data sources were finally included. Although there is a section of the paper titled "Characteristics of included studies"; this section only describes some features of studies that were included, it does not expressly and explicitly state why they were included; ie. inclusion criteria for inclusion of studies is not clearly stated. There is no section of the document that describes or categorically states the exclusion criteria that were used to make decisions to exclude those studies that were excluded; i.e. exclusion criteria for exclusion of studies is also not clearly stated. Would it not be better if there could be a section of the document that clearly and explicitly states and provides the inclusion and exclusion criteria for selecting studies/data sources for this work?

Response: Thank you for your comment. A clear and explicit description of the inclusion and exclusion criteria has been added in the “Method” section under “Eligibility” as follow:

Eligibility

Inclusion criteria

All cross-sectional, population-based and cohort studies that investigated the prevalence of male circumcision in sub-Saharan Africa were eligible for inclusion in this review. The eligible studies should have been published during the period 1 January 2010 and 7 April 2023. This period 2010-2023 was selected as most VMMC programs started either just before or during the year 2010 (in the case of South Africa) after the WHO recommendation in 2007 [14]. No language restriction was applied, and we did not attempt to distinguish between VMMC and traditional and cultural male circumcision as many data sources did not do separate analyses by type of male circumcision and others did not specify the type of circumcision. 

Exclusion criteria

Duplicate data sources, studies that did not report male circumcision prevalence by country in the analysis, studies that were published before or after the period of interest and studies that included data for countries outside the 15 priority countries were excluded.

---

## [Editor Report · Decision Letter 1]

24 Jan 2024

Estimates of the prevalence of male circumcision in Sub-Saharan Africa from 2010 – 2023 – a systematic review and meta-analysis

PONE-D-23-37643R1

Dear Tawanda Chivese,

We’re pleased to inform you that your manuscript has been judged scientifically suitable for publication and will be formally accepted for publication once it meets all outstanding technical requirements.

Kind regards,

Taurayi A Tafuma, MPH, MBChB

Academic Editor

PLOS ONE
---

## [Editor Report · Acceptance letter]

4 Mar 2024

PONE-D-23-37643R1 

PLOS ONE

Dear Dr. Chivese, 

I'm pleased to inform you that your manuscript has been deemed suitable for publication in PLOS ONE. Congratulations! Your manuscript is now being handed over to our production team.

Kind regards, 

on behalf of

Dr. Taurayi A Tafuma 

Academic Editor

PLOS ONE